# AUTONOMOUS REINFORCEMENT LEARNING: FORMALISM AND BENCHMARKING

**Archit Sharma**[*1]     **Kelvin Xu**[*2]     **Nikhil Sardana**[1]
**Abhishek Gupta**[3]     **Karol Hausman**[4]     **Sergey Levine**[2]     **Chelsea Finn**[1]
[1]Stanford University
[2]University of California, Berkeley
[3]MIT
[4]Google Brain

## ABSTRACT

Reinforcement learning (RL) provides a naturalistic framing for learning through trial and error, which is appealing both because of its simplicity and effectiveness and because of its resemblance to how humans and animals acquire skills through experience. However, real-world embodied learning, such as that performed by humans and animals, is situated in a continual, non-episodic world, whereas common benchmark tasks in RL are episodic, with the environment resetting between trials to provide the agent with multiple attempts. This discrepancy presents a major challenge when attempting to take RL algorithms developed for episodic simulated environments and run them on real-world platforms, such as robots. In this paper, we aim to address this discrepancy by laying out a framework for *Autonomous Reinforcement Learning* (ARL): reinforcement learning where the agent not only learns through its own experience, but also contends with lack of human supervision to reset between trials. We introduce a simulated benchmark *EARL*[1] around this framework, containing a set of diverse and challenging simulated tasks reflective of the hurdles introduced to learning when only a minimal reliance on extrinsic intervention can be assumed. We show that standard approaches to episodic RL and existing approaches struggle as interventions are minimized, underscoring the need for developing new algorithms for reinforcement learning with a greater focus on autonomy.

## 1 INTRODUCTION

One of the appeals of reinforcement learning is that it provides a naturalistic account for how complex behavior could emerge autonomously from trial and error, similar to how humans and animals can acquire skills through experience in the real world. Real world learning however, is situated in a continual, non-episodic world, whereas commonly benchmarks in RL often assume access to an oracle reset mechanism that provides agents the ability to make multiple attempts. This presents a major challenge with attempting to deploy RL algorithms in environments where such an assumption would necessitate costly human intervention or manual engineering. Consider an example of a robot learning to clean and organize a home. We would ideally like the robot to be able to autonomously explore the house, understand cleaning implements, identify good strategies on its own throughout this process, and adapt when the home changes. This vision of robotic learning in real world, continual, non-episodic manner is in stark contrast to typical experimentation in reinforcement learning in the literature (Levine et al., 2016; Chebotar et al., 2017; Yahya et al., 2017; Ghadirzadeh et al., 2017; Zeng et al., 2020), where agents must be consistently reset to a set of initial conditions through human effort or engineering so that they may try again. Autonomy, especially for data hungry approaches such as reinforcement learning, is an enabler of broad applicability, but is rarely an explicit consideration. In this work, we propose to bridge this specific gap by providing a formalism and set of benchmark tasks that considers the challenges faced by agents situated in non-episodic environment, rather than treating them as being abstracted away by an oracle reset.

---

[*]Equal contribution. Corresponding authors: architsh@stanford.edu, kelvinxu@berkeley.edu

[1]Code and related information for EARL can be found at architsharma97.github.io/earl_benchmark/

Our specific aim is to place a greater focus on developing algorithms under assumptions that more closely resemble autonomous real world robotic deployment. While the episodic setting naturally captures the notion of completing a "task", it hides the costs of assuming oracle resets, which when removed can cause algorithms developed in the episodic setting to perform poorly (Sec. 6.1). Moreover, while prior work has examined settings such as RL without resets (Eysenbach et al., 2017; Xu et al., 2020; Zhu et al., 2020; Gupta et al., 2021), ecological RL (Co-Reyes et al., 2020), or RL amidst non-stationarity (Xie et al., 2020) in isolated scenarios, these settings are not well-represented in existing benchmarks. As a result, there is not a consistent formal framework for evaluating autonomy in reinforcement learning and there is limited work in this direction compared to the vast literature on reinforcement learning. By establishing this framework and benchmark, we aim to solidify the importance of algorithms that operate with greater autonomy in RL.

Before we can formulate a set of benchmarks for this type of autonomous reinforcement learning, we first formally define the problem we are solving. As we will discuss in Section 4, we can formulate two distinct problem settings where autonomous learning presents a major challenge. The first is a setting where the agent first trains in a non-episodic environment, and is then "deployed" into an episodic test environment. In this setting, which is most commonly studied in prior works on "reset-free" learning (Han et al., 2015; Zhu et al., 2020; Sharma et al., 2021), the goal is to learn the best possible *episodic* policy after a period of non-episodic training. For instance, in the case of a home cleaning robot, this would correspond to evaluating its ability to clean a messy home. The second setting is a continued learning setting: like the first setting, the goal is to learn in a non-episodic environment, but there is no distinct "deployment" phase, and instead the agent must minimize regret over the duration of the training process. In the previous setting of the home cleaning robot, this would evaluate the persistent cleanliness of the home. We discuss in our autonomous RL problem definition how these settings present a number of unique challenges, such as challenging exploration.

The main contributions of our work consist of a benchmark for autonomous RL (ARL), as well as formal definitions of two distinct ARL settings. Our benchmarks combine components from previously proposed environments (Coumans & Bai, 2016; Gupta et al., 2019; Yu et al., 2020; Gupta et al., 2021; Sharma et al., 2021), but reformulate the learning tasks to reflect ARL constraints, such as the absence of explicitly available resets. Our formalization of ARL relates it to the standard RL problem statement, provides a concrete and general definition, and provides a number of instantiations that describe how common ingredients of ARL, such as irreversible states, interventions, and other components, fit into the general framework. We additionally evaluate a range of previously proposed algorithms on our benchmark, focusing on methods that explicitly tackle reset-free learning and other related scenarios. We find that both standard RL methods and methods designed for reset-free learning struggle to solve the problems in the benchmark and often get stuck in parts of the state space, underscoring the need for algorithms that can learn with greater autonomy and suggesting a path towards the development of such methods.

## 2 RELATED WORK

Prior work has proposed a number of benchmarks for reinforcement learning, which are often either explicitly episodic (Todorov et al., 2012; Beattie et al., 2016; Chevalier-Boisvert et al., 2018), or consist of games that are implicitly episodic after the player dies or completes the game (Bellemare et al., 2013; Silver et al., 2016). In addition, RL benchmarks have been proposed in the episodic setting for studying a number of orthogonal questions, such multi-task learning (Bellemare et al., 2013; Yu et al., 2020), sequential task learning (Wołczyk et al., 2021), generalization (Cobbe et al., 2020), and multi-agent learning (Samvelyan et al., 2019; Wang et al., 2020). These benchmarks differ from our own in that we propose to study the challenge of autonomy. Among recent benchmarks, the closest to our own is Jelly Bean World (Platanios et al., 2020), which consists of a set of procedural generated gridworld tasks. While this benchmark also considers the non-episodic setting, our work is inspired by the challenge of autonomous learning in robotics, and hence considers an array of manipulation and locomotion tasks. In addition, our work aims to establish a conceptual framework for evaluating prior algorithms in light of the requirement for persistent autonomy.

Enabling embodied agents to learn continually with minimal interventions is a motivation shared by several subtopics of reinforcement learning research. The setting we study in our work shares conceptual similarities with prior work in continual and lifelong learning (Schmidhuber, 1987; Thrun & Mitchell, 1995; Parisi et al., 2019; Hadsell et al., 2020). In context of reinforcement learning, this work has studied the problem of episodic learning in sequential MDPs (Khetarpal et al., 2020; Rusu

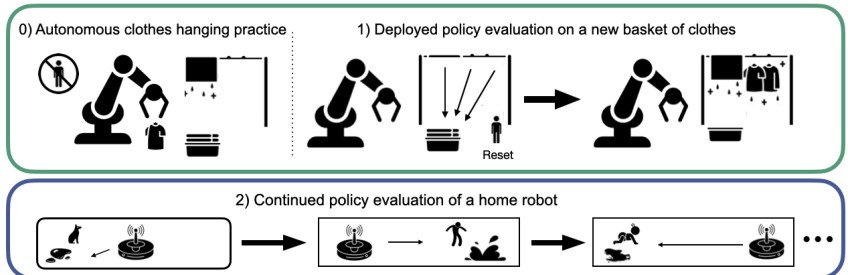

Figure 1: Two evaluation schemes in autonomous RL. First, the deployment setting (top row, (1)), where we are interested in obtaining a policy during a training phase, $\pi$, that performs well when deployed from a $s_0 \sim \rho$. Second, the continuing setting (bottom row, (2)), where a floor cleaning robot is tasked with keeping a floor clean and is only evaluated on its cumulative performance (Eq. 2) over the agent's lifetime.

et al., 2016; Kirkpatrick et al., 2017; Fernando et al., 2017; Schwarz et al., 2018; Mendez et al., 2020), where the main objective is forward/backward transfer or learning in non-stationary dynamics (Chandak et al., 2020; Xie et al., 2020; Lu et al., 2020). In contrast, the emphasis of our work is learning in non-episodic settings while literature in continual RL assumes an episodic setting. As we will discuss, learning autonomously without access to oracle resets is a hard problem even when the task-distribution and dynamics are stationary. In a similar vein, unsupervised RL (Gregor et al., 2016; Pong et al., 2019; Eysenbach et al., 2018; Sharma et al., 2019; Campos et al., 2020) also enables skill acquisition in the absence of rewards, reducing human intervention required for designing reward functions. These works are complimentary to our proposal and form interesting future work.

Reset-free RL has been studied by previous works with a focus on safety (Eysenbach et al., 2017), automated and unattended learning in the real world (Han et al., 2015; Zhu et al., 2020; Gupta et al., 2021), skill discovery (Xu et al., 2020; Lu et al., 2020), and providing a curriculum (Sharma et al., 2021). Strategies to learn reset-free behavior include directly learning a backward reset controller (Eysenbach et al., 2017), learning a set of auxillary tasks that can serve as an approximate reset (Ha et al., 2020; Gupta et al., 2021), or using a novelty seeking reset controller (Zhu et al., 2020). Complementary to this literature, we aim to develop a set of benchmarks and a framework that allows for this class of algorithms to be studied in a unified way. Instead of proposing new algorithms, our work is focused on developing a set of unified tasks that emphasize and allow us to study algorithms through the lens of autonomy.

## 3 PRELIMINARIES

Consider a Markov Decision Process (MDP) $\mathcal{M} \equiv (\mathcal{S}, \mathcal{A}, p, r, \rho, \gamma)$ (Sutton & Barto, 2018). Here, $\mathcal{S}$ denotes the state space, $\mathcal{A}$ denotes the action space, $p : \mathcal{S} \times \mathcal{A} \times \mathcal{S} \mapsto \mathbb{R}_{\geq 0}$ denotes the transition dynamics, $r : \mathcal{S} \times \mathcal{A} \mapsto \mathbb{R}$ denotes the reward function, $\rho : \mathcal{S} \mapsto \mathbb{R}_{\geq 0}$ denotes the initial state distribution and $\gamma \in [0, 1)$ denotes the discount factor. The objective in reinforcement learning is to maximize $J(\pi) = \mathbb{E}[\sum_{t=0}^{\infty} \gamma^t r(s_t, a_t)]$ with respect to the policy $\pi$, where $s_0 \sim \rho(\cdot)$, $a_t \sim \pi(\cdot \mid s_t)$ and $s_{t+1} \sim p(\cdot \mid s_t, a_t)$. Importantly, the RL framework assumes the ability to sample $s_0 \sim \rho$ arbitrarily. Typical implementations of reinforcement learning algorithms carry out thousands or millions of these trials, implicitly requiring the environment to provide a mechanism to be "reset" to a state $s_0 \sim \rho$ for every trial.

## 4 AUTONOMOUS REINFORCEMENT LEARNING

In this section, we develop a framework for autonomous reinforcement learning (ARL) that formalizes reinforcement learning in settings without extrinsic interventions. We first define a non-episodic training environment where the agent can autonomously interact with its environment in Section 4.1, building on the formalism of standard reinforcement learning. We introduce two distinct evaluation settings as visualized in Figure 1: Section 4.1.1 discusses the *deployment setting* where the agent will be deployed in a test environment after training, and the goal is to maximize this "deployed" performance. Section 4.1.2 discusses the *continuing setting*, where the agent has no separate deployment phase and aims to maximize the reward accumulated over its lifetime. In its most general form, the latter corresponds closely to standard RL, while the former can be interpreted as a kind of transfer learning. As we will discuss in Section 4.2, this general framework can be instantiated such

that, for different choices of the underlying MDP, we can model different realistic autonomous RL scenarios such as settings where a robot must learn to reset itself between trials or settings with non-reversible dynamics. Finally, Section 4.3 considers algorithm design for autonomous RL, discussing the challenges in autonomous operation while also contrasting the evaluation protocols.

## 4.1 GENERAL SETUP

Our goal is to formalize a problem setting for autonomous reinforcement learning that encapsulates realistic autonomous learning scenarios. We define the setup in terms of a training MDP $\mathcal{M}_T \equiv (\mathcal{S}, \mathcal{A}, p, r, \rho)$, where the environment initializes to $s_0 \sim \rho$, and then the agent interacts with the environment autonomously from then on. Note, this lack of episodic resets in our setup departs not only from the standard RL setting, but from other continual reinforcement learning settings e.g. Wołczyk et al. (2021), where resets are provided between tasks. Symbols retain their meaning from Section 3. In this setting, a *learning algorithm* $\mathbb{A}$ can be defined as a function $\mathbb{A} : \{s_i, a_i, s_{i+1}, r_i\}_{i=0}^{t-1} \mapsto (a_t, \pi_t)$, which maps the transitions collected in the environment until the time $t$ (e.g., a replay buffer), to a (potentially exploratory) action $a_t \in \mathcal{A}$ applied in the environment and its best guess at the optimal policy $\pi_t : \mathcal{S} \times \mathcal{A} \mapsto \mathbb{R}_{\geq 0}$ used for *evaluation* at time $t$. We note that the assumption of a reward function implicitly requires human engineering, but in principle could be relaxed by methods that learn reward functions from data. In addition, we note that $a_t$ does not need to come from $\pi_t$, which is already implicit in most reinforcement learning algorithms: $Q$-learning (Sutton & Barto, 2018) methods such as DQN (Mnih et al., 2015), DDPG (Lillicrap et al., 2015) use an $\epsilon$-greedy policy as an exploration policy on top of the greedy policy for evaluation. However, our setting necessitates more concerted exploration, and the exploratory action may come from an entirely different policy. Note that the initial state distribution is sampled ($s_0 \sim \rho$) exactly *once* to begin training, and then the algorithm $\mathbb{A}$ is run until $t \to \infty$, generating the sequence $s_0, a_0, s_1, a_1, \ldots$ in the MDP $\mathcal{M}_T$. This is the primary difference compared to the episodic setting described in Section 3, which can sample the initial state distribution repeatedly.

### 4.1.1 ARL DEPLOYMENT SETTING

Consider the problem where a robot has to learn how to close a door. Traditional reinforcement learning algorithms require several trials, repeatedly requiring interventions to open the door between trials. The desire is that the robot autonomously interacts with the door, learning to open it if it is required to practice closing the door. The output policy of the training procedure is evaluated in its deployment setting, in this case on its ability to close the door. Formally, the evaluation objective $J_D$ for a policy $\pi$ in the deployment setting can be written as:

$$J_D(\pi) = \mathbb{E}_{s_0 \sim \rho, a_j \sim \pi(\cdot|s_j), s_{j+1} \sim p(\cdot|s_j, a_j)} \Big[ \sum_{j=0}^{\infty} \gamma^j r(s_j, a_j) \Big]. \tag{1}$$

**Definition 1** (*Deployed Policy Evaluation*). For an algorithm $\mathbb{A} : \{s_i, a_i, s_{i+1}\}_{i=0}^{t-1} \mapsto (a_t, \pi_t)$, deployed policy evaluation $\mathbb{D}(\mathbb{A})$ is given by $\mathbb{D}(\mathbb{A}) = \sum_{t=0}^{\infty} (J_D(\pi^*) - J_D(\pi_t))$, where $J_D(\pi)$ is defined in Eq 1 and $\pi^* \in \arg\max_\pi J_D(\pi)$.

The evaluation objective $J_D(\pi)$ is identical to the one defined in Section 3 on MDP $\mathcal{M}$ (deployment environment). The policy evaluation is "hypothetical", the environment rollouts used for evaluating policies are not used in training. Even though the evaluation trajectories are rolled out from the initial state, there are no interventions in training. Concretely, the algorithmic goal in this setting can be stated as $\min_{\mathbb{A}} \mathbb{D}(\mathbb{A})$. In essence, the policy outputs $\pi_t$ from the autonomous algorithm $\mathbb{A}$ should match the oracle *deployment* performance, i.e. $J_D(\pi^*)$, as quickly as possible. Note that $J_D(\pi^*)$ is a constant that can be ignored when comparing two algorithms, i.e. we only need to know $J_D(\pi_t)$ for a given algorithm in practice.

### 4.1.2 ARL CONTINUING SETTING

For some applications, the agent's experience cannot be separated into training and deployment phases. Agents may have to learn and improve in the environment that they are "deployed" into, and thus these algorithms need to be evaluated on their performance during the agent's lifetime. For example, a robot tasked with keeping the home clean learns and improves on the job as it adapts to the home in which it is deployed. To this end, consider the following definition:

**Definition 2** (*Continuing Policy Evaluation*). For an algorithm $\mathbb{A} : \{s_i, a_i, s_{t+1}\}_{i=0}^{t-1} \mapsto (a_t, \pi_t)$, continuing policy evaluation $\mathbb{C}(\mathbb{A})$ can be defined as:

$$\mathbb{C}(\mathbb{A}) = \lim_{h \to \infty} \frac{1}{h} \mathbb{E}_{s_0 \sim \rho, a_t \sim \mathbb{A}(\{s_i, a_i, s_{i+1}\}_{i=0}^{t-1}), s_{t+1} \sim p(\cdot | s_t, a_t)} \Big[ \sum_{t=0}^{h} r(s_t, a_t) \Big] \tag{2}$$

Here, $a_t$ is the action taken by the algorithm $\mathbb{A}$ based on the transitions collected in the environment until time $t$, measuring the performance under reward $r$. The optimization objective can be stated as $\max_{\mathbb{A}} \mathbb{C}(\mathbb{A})$. Note that $\pi_t$ is not used in computing $\mathbb{C}(\mathbb{A})$. In practice, this amounts to measuring the reward collected by the agent in the MDP $\mathcal{M}_T$ during its lifetime[2].

### 4.2 HOW SPECIFIC ARL PROBLEMS FIT INTO OUR FRAMEWORK

The framework can easily be adapted to model possible autonomous reinforcement learning scenarios that may be encountered:

**Intermittent interventions.** By default, the agent collects experience in the environment with fully autonomous interaction in the MDP $\mathcal{M}_T$. However, we can model the occasional intervention with transition dynamics defined as $\tilde{p}(\cdot \mid s, a) = (1 - \epsilon)p(\cdot \mid s, a) + \epsilon\rho(\cdot)$, where the next state is sampled with $1 - \epsilon$ probability from the environment dynamics or with $\epsilon$ probability from the initial state distribution via intervention for some $\epsilon \in [0, 1]$. A low $\epsilon$ represents very occasional interventions through the training in MDP $\mathcal{M}_T$. In fact, the framework described in Section 3, which is predominantly assumed by reinforcement learning algorithms, can be understood to have a large $\epsilon$. To contextualize $\epsilon$, the agent should expect to get an intervention after $1/\epsilon$ steps in the environment. Current episodic settings typically provide an environment reset every 100 to 1000 steps, corresponding to $\epsilon \in (1e\text{-}3, 1e\text{-}2)$ and an autonomous operation time of typically a few seconds to few minutes depending on the environment. While full autonomy would be desirable (that is, $\epsilon = 0$), intervening every few hours to few days may be reasonable to arrange, which corresponds to environment resets every 100,000 to 1,000,000 steps or $\epsilon \in (1e\text{-}6, 1e\text{-}5)$. We evaluate the dependence of algorithms designed for episodic reinforcement learning on the reset frequency in Section 6.1.

**Irreversible states.** An important consideration for developing autonomous algorithms is the "reversibility" of the underlying MDP $\mathcal{M}$. Informally, if the agent can reverse any transition in the environment, the agent is guaranteed to not get stuck in the environment. As an example, a static robot arm can be setup such that there always exists an action sequence to open or close the door. However, the robot arm can push the object out of its reach such that no action sequence can retrieve it. Formally, we require MDPs to be ergodic for them to be considered reversible (Moldovan & Abbeel, 2012). In the case of non-ergodic MDPs, we adapt the ARL framework to enable the agent to request extrinsic interventions, which we discuss in Appendix A.

### 4.3 DISCUSSION OF THE ARL FORMALISM

The ARL framework provides two evaluation protocols for autonomous RL algorithms. Algorithms can typically optimize only one of the two evaluation metrics. *Which evaluation protocol should the designer optimize for?* In a sense, the need for two evaluation protocols arises from task specific constraints, which themselves can be sometimes relaxed depending on the specific trade-off between the cost of real world training and the cost of intervention. The continuing policy evaluation represents the oracle metric one should strive to optimize when continually operational agents are deployed into dynamic environments. The need for *deployment policy evaluation* arises from two implicit practical constraints: (a) requirement of a large number of trials to solve desired tasks and (b) absence of interventions to enable those trials. If either of these can be easily relaxed, then one could consider optimizing for *continuing policy evaluation*. For example, if the agent can learn in a few trials because it was meta-trained for quick adaptation (Finn et al., 2017), providing a few interventions for those trials may be reasonable. Similarly, if the interventions are easy to obtain during deployment without incurring significant human cost, perhaps through scripted behaviors or enabled by the deployment setting (for example, sorting trash in a facility), the agent can repeatedly try the task and learn while deployed. However, if these constraints cannot be relaxed at deployment, one should consider optimizing for the *deployment policy evaluation* since this incentivizes the agents to learn targeted behaviors by setting up its own practice problems.

---

[2]We can also consider the more commonly used setting of expected discounted sum of rewards as the objective. To ensure that future rewards are relevant, the discount factors would need to be much larger than values typically used.

## 5   EARL: ENVIRONMENTS FOR AUTONOMOUS REINFORCEMENT LEARNING

In this section, we introduce the set of environments in our proposed benchmark, **E**nvironments for **A**utonomous **R**einforcement **L**earning (EARL). We first discuss the factors in our design criteria and provide a description of how each environment fits into our overall benchmark philosophy, before presenting the results and analysis. For detailed descriptions of each environment, see Appendix A.

### 5.1   BENCHMARK DESIGN FACTORS

**Representative Autonomous Settings.**  We include a broad array of tasks that reflect the types of autonomous learning scenarios agents may encounter in the real world. This includes different problems in manipulation and locomotion, and tasks with multiple object interactions for which it would be challenging to instrument resets. We also ensure that both the continuing and deployment evaluation protocols of ARL are realistic representative evaluations.

**Directed Exploration**. In the autonomous setting, the necessity to practice a task again, potentially from different initial states, gives rise to the need for agents to learn rich reset behaviors. For example, in the instance of a robot learning to interact with multiple objects in a kitchen, the robot must also learn to implicitly or explicitly compose different reset behaviors.

**Rewards and Demonstrations**. One final design aspect for our benchmark is the choice of reward functions. Dense rewards are a natural choice in certain domains (e.g., locomotion), but designing and providing dense rewards in real world manipulation domains can be exceptionally challenging. Sparse rewards are easier to specify in such scenarios, but this often makes exploration impractical. As a result, prior work has often leveraged demonstrations (e.g., (Gupta et al., 2019)), especially in real world experimentation. To reflect practical usage of RL in real world manipulation settings, we include a small number of demonstrations for the sparse-reward manipulation tasks.

### 5.2   ENVIRONMENT DESCRIPTIONS

**Tabletop-Organization (TO).** The Tabletop-Organization task is a diagnostic object manipulation environment proposed by Sharma et al. (2021).  The agent consists of a gripper modeled as a pointmass, which can grasp objects that are close to it.  The agent's goal is to bring a mug to four different locations designated by a goal coaster. The agent's reward function is a sparse indicator function when the mug is placed at the goal location.    Limited demonstrations are provided to the agent.

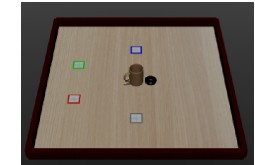

**Sawyer-Door (SD).**   The Sawyer-Door task, from the MetaWorld benchmark (Yu et al., 2020) consists of a Sawyer robot arm who's goal is to close the door whenever it is in an open position.  The task reward is a sparse indicator function based on the angle of the door.  Repeatedly practicing this task implicitly requires the agent to learn to open the door.  Limited demonstrations for opening and closing the door are provided.

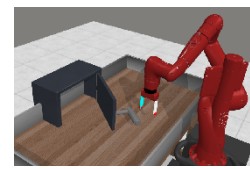

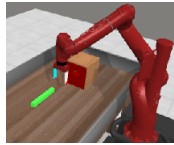

**Sawyer-Peg (SP).**   The Sawyer-Peg task (Yu et al., 2020) consists of a Sawyer robot required to insert a peg into a designed goal location.  The task reward is a sparse indicator function for when the peg is in the goal location.  In the deployment setting, the agent must learn to insert the peg starting on the table. Limited demonstrations for inserting and removing the peg are provided.

**Franka-Kitchen (FK).**   The Franka-Kitchen (Gupta et al., 2019) is a domain where a 9-DoF robot, situated in a kitchen environment, is required to solve tasks consisting of compound object interactions. The environment consists of a microwave, a hinged cabinet, a burner, and a slide cabinet. One example task is to open the microwave, door and burner. This domain presents a number of distinct challenges for ARL. First, the compound nature of each

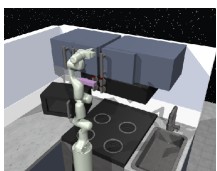

task results in a challenging long horizon problem, which introduces exploration and credit assignment challenges. Second, while generalization is important in solving the environment, combining *reset* behaviors are equally important given the compositional nature of the task.

**DHand-LightBulb (DL).** The DHand-Lightbulb environment consists of a 22-DoF 4 fingered hand, mounted on a 6 DoF Sawyer robot. The environment is based on one originally proposed by Gupta et al. (2021). The task in this domain is for the robot to grasp pickup a lightbulb to a specific location. The high-dimensional action space makes the task extremely challenging. In the deployment setting, the bulb can be initialized anywhere on the table, testing the agent on a wide initial state distribution. 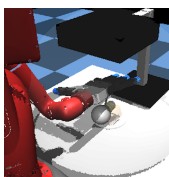

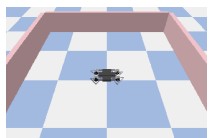 **Minitaur-Pen (MP).** Finally, the Minitaur-Pen task consists of an 8-DoF Minitaur robot (Coumans & Bai, 2016) confined to a pen environment. The goal of the agent is to navigate to a set of goal locations in the pen. The task is designed to mimic the setup of leaving a robot to learn to navigate within an enclosed setting in an autonomous fashion. This task is different from the other tasks given it is a locomotion task, as opposed to the other tasks being manipulation tasks.

## 6 BENCHMARKING AND ANALYSIS

The aim of this section is to understand the challenges in autonomous reinforcement learning and to evaluate the performance and shortcomings of current autonomous RL algorithms. In Section 6.1, we first evaluate standard episodic RL algorithms in ARL settings as they are required to operate with increased autonomy, underscoring the need for a greater focus on autonomy in RL algorithms. We then evaluate prior autonomous learning algorithms on EARL in Section 6.2. While these algorithms do improve upon episodic RL methods, they fail to make progress on more challenging tasks compared to methods provided with oracle resets leaving a large gap for improvement. Lastly, in Section 6.3, we investigate the learning of existing algorithms, providing a hypothesis for their inadequate performance. We also find that when autonomous RL does succeed, it tends to find more robust policies, suggesting an intriguing connection between autonomy and robustness.

### 6.1 GOING FROM STANDARD RL TO AUTONOMOUS RL

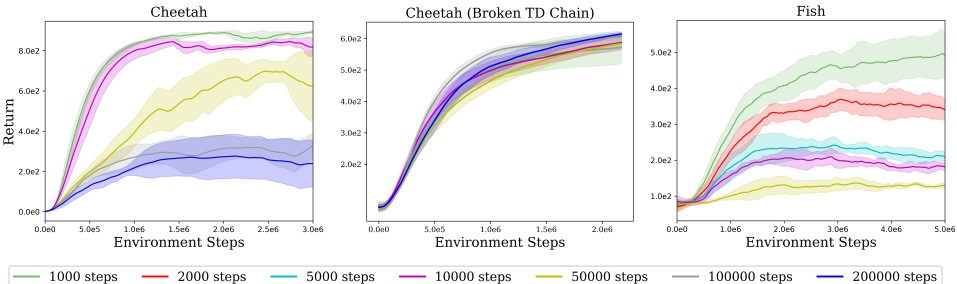

Figure 2: Performance of standard RL at with varying levels of autonomy, ranging from resets provided every 1000 to 200000 steps. Performance degrades substantially as environment resets become infrequent.

In this section, we evaluate standard RL methods to understand how their performance changes when they are applied naïvely to ARL problems. To create a continuum, we will vary the level of "autonomy" (i.e., frequency of resets), corresponding to $\epsilon$ as defined in Section 4.2. For these experiments only, we use the simple `cheetah` and `fish` environments from the DeepMind Control Suite (Tassa et al., 2018). We use soft actor-critic (SAC) (Haarnoja et al., 2018a) as a representative standard RL algorithm. We consider different training environments with increasing number of steps between resets, ranging from 1000 to 200,000 steps. Figure 2 shows the performance of the learned policy as the training progresses, where the return is measured by running the policy for 1000 steps. The `cheetah` environment is an infinite-horizon running environment, so changing the training horizon should not affect the performance theoretically. However, we find that the performance degrades drastically as the training horizon is increased, as shown in Fig 2 (*left*). We attribute this problem to a combination of function approximation and temporal difference learning. Increasing the episode length destabilizes the learning as the effective bootstrapping length increases: the $Q$-value function $Q^\pi(s_0, a_0)$ bootstraps on the value of $Q^\pi(s_1, a_1)$, which bootstraps on $Q^\pi(s_2, a_2)$ and so on till $Q^\pi(s_{100,000}, a_{100,000})$. To break this chain, we consider a biased TD update: $Q^\pi(s_t, a_t) \leftarrow r(s_t, a_t) + \gamma Q^\pi(s_{t+1}, a_{t+1})$ if $t$ is not a multiple of 1000, else

| Method | TO | SD | SP | FK | DL | MP |
|--------|-----|-----|-----|-----|-----|-----|
| *naïve RL* | 0.32 (0.17) | 0.00 (0.00) | 0.00 (0.00) | -2705.21 (167.10) | -239.30 (8.85) | -1041.10 (44.58) |
| *FBRL* | 0.94 (0.04) | **1.00 (0.00)** | 0.00 (0.00) | -2733.15 (324.10) | -242.38 (8.84) | -986.34 (67.95) |
| *R3L* | 0.96 (0.04) | 0.54 (0.18) | 0.00 (0.00) | -2639.28 (233.28) | 728.54 (122.86) | -186.30 (34.79) |
| *VaPRL* | **0.98 (0.02)** | 0.94 (0.05) | 0.02 (0.02) | - | - | - |
| *oracle RL* | 0.80 (0.11) | **1.00 (0.00)** | **1.00 (0.00)** | 1203.88 (203.86) | 2028.75 (35.95) | **-41.50 (3.40)** |

Table 1: Average return of the final deployed policy. Performance is averaged over 5 random seeds. The mean and and the standard error are reported, with the best performing entry in bold. For sparse reward domains (**TO, SD, SP**), 1.0 indicates the maximum performance and 0.0 indicates minimum performance.

| Method | TO | SD | SP | FK | DL | MP |
|--------|-----|-----|-----|-----|-----|-----|
| *naïve RL* | **0.012 (0.004)** | 0.373 (0.073) | $< 0.001$ | **-4.944 (0.440)** | -0.734 (0.024) | -18.193 (2.375) |
| *FBRL* | 0.005 (0.001) | 0.329 (0.080) | **0.003 (0.003)** | -8.754 (0.405) | -0.747 (0.014) | -22.087 (1.285) |
| *R3L* | 0.001 (0.000) | 0.369 (0.016) | $< 0.001$ | -6.577 (0.309) | **0.091 (0.026)** | **-1.093 (0.020)** |
| *VaPRL* | 0.009 (0.000) | **0.574 (0.085)** | $< 0.001$ | - | - | - |

Table 2: Average reward accumulated over the training lifetime in accordance to continuing policy evaluation. Performance is averaged over 5 random seeds. The mean and the standard error (brackets) are reported, with the best performing entry in bold.

$Q^\pi(s_t, a_t) \leftarrow r(s_t, a_t)$. This is inspired by practical implementations of SAC (Haarnoja et al., 2018b), where the $Q$-value function regresses to $r(s, a)$ for terminal transitions to stabilize the training. This effectively fixes the problem for `cheetah`, as shown in Figure 2 (*middle*). However, this solution does not translate in general, as can be seen observed in the `fish` environment, where the performance continues to degrade with increasing training horizon, shown in Fig 2 (*right*). The primary difference between `cheetah` and `fish` is that the latter is a goal-reaching domain. The `cheetah` can continue improving its gait on the infinite plane without resets, whereas the `fish` needs to undo the task to practice goal reaching again, creating a non-trivial exploration problem.

## 6.2 EVALUATION: SETUP, METRICS, BASELINES, AND RESULTS

**Training Setup.** Every algorithm $\mathbb{A}$ receives a set of initial states $s_0 \sim \rho$, and a set of goals from the goal-distribution $g \sim p(g)$. Demonstrations, if available, are also provided to $\mathbb{A}$. We consider the *intermittent interventions* scenario described in Section 4.2 for the benchmark. In practice, we reset the agent after a fixed number of steps $H_T$. The value of $H_T$ is fixed for every environment, ranging between $100,000$ - $400,000$ steps. Every algorithm $\mathbb{A}$ is run for a fixed number of steps $H_{max}$ after which the training is terminated. Environment-specific details are in Appendix A.4.

**Evaluation Metrics.** For deployed policy evaluation, we compute $\mathbb{D}(\mathbb{A}) = -\sum_{t=0}^{H_{max}} J_D(\pi_t)$, ignoring $J_D(\pi^*)$ as it is a constant for all algorithms. Policy evaluation $J_D(\pi_t)$ is carried out every 10000 training steps, where $J_D(\pi_t) = \sum_{t=0}^{H_E} r(s_t, a_t)$ is the average return accumulated over an episode of $H_E$ steps by running the policy $\pi_t$ 10 times, starting from $s_0 \sim \rho$ for every trial. These roll-outs are only used for evaluation, and are not provided to the algorithm. In practice, we plot $J_D(\pi_t)$ versus time $t$, such that minimizing $\mathbb{D}(\mathbb{A})$ can be understood as maximizing the area under the learning curve, which we find more interpretable. Given a finite training budget $H_{max}$, the policy $\pi_t$ may be quite suboptimal compared to $\pi^*$. Thus, we also report the performance of the final policy, that is $J_D(\pi_{H_{max}})$ in Table 1. For continuing policy evaluation $\mathbb{C}(\mathbb{A})$, we compute the average reward as $\bar{r}(h) = \sum_{t=0}^{h} r(s_t, a_t)/h$. We plot $\bar{r}(h)$ versus $h$, while we report $\bar{r}(H_{max})$ in Table 2. The evaluation curves corresponding to continuing and deployed policy evaluation are in Appendix A.6.

**Baselines.** We evaluate forward-backward RL (**FBRL**) (Han et al., 2015; Eysenbach et al., 2017), a perturbation controller (**R3L**) (Zhu et al., 2020), value-accelerated persistent RL (**VaPRL**) (Sharma et al., 2021), a comparison to simply running the base RL algorithm with the biased TD update discussed in Section 6.1 (**naïve RL**), and finally an oracle (**oracle RL**) where resets are provided are provided every $H_E$ steps ($H_T$ is typically three orders of magnitude larger than $H_E$). We benchmark VaPRL only when demonstrations are available, in accordance to the proposed algorithm in Sharma et al. (2021). We average the performance of all algorithms across 5 random seeds. More details pertaining to these algorithms can be found in Appendix A.3, A.5.

Overall, we see in Table 1 that based on the deployed performance of the final policy, autonomous RL algorithms substantially underperform oracle RL and fail to make any progress on `sawyer-peg` and `franka-kitchen`. Notable exceptions are the performances of VaPRL on

`tabletop-organization` and R3L on `minitaur-pen`, outperforming oracle RL. Amongst the autonomous RL algorithms, VaPRL does best when the demonstrations are given and R3L does well on domains when no demonstrations are provided. Nonetheless, this leaves substantial room for improvement for future works evaluating on this benchmark. More detailed learning curves are shown in Section A.6. In the continuing setting, we find that naïve RL performs well on certain domains (best on 2 out of 6 domains). This is unsurprising, as naïve RL is incentivized to occupy the final "goal" position, and continue to accumulate over the course of its lifetime, whereas other algorithms are explicitly incentivized to explore. Perhaps surprisingly, we find that VaPRL on `sawyer-door` and R3L on `dhand-lightbulb` and `minitaur` does better than naïve RL, suggesting that optimizing for deployed performance can also improve the continuing performance. Overall, we find that performance in the continuing setting does not necessarily translate to improved performance in the deployed policy evaluation, emphasizing the differences between these two evaluation schemes.

## 6.3 ANALYZING AUTONOMOUS RL ALGORITHMS

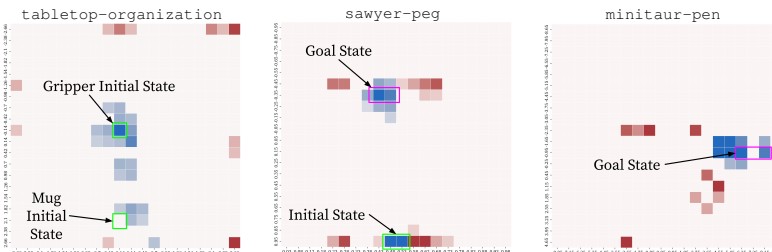

Figure 3: Comparing the distribution of states visited with resets (*blue*) and without resets (*brown*). Heatmaps visualize the difference between state visitations for oracle RL and FBRL, thresholded to highlight states with large differences. Resets enable the agent to stay around the initial state distribution and the goal distribution, whereas the agents operating autonomously skew farther away, posing an exploration challenge.

A hypothesis to account for the relative underperformance of autonomous RL algorithms compared to oracle RL is that environment resets constrain the state distribution visited by the agent close to the initial and the goal states. When operating autonomously for long periods of time, the agent can skew far away from the goal states, creating a hard exploration challenge. To test this hypothesis, we compare the state distribution when using oracle RL (*in blue*) versus FBRL (*in brown*) in Figure 3. We visualize the $(x, y)$ positions visited by the gripper for `tabletop-organization`, the $(x, y)$ positions of the peg for `sawyer-peg` and the $x, y$ positions of the minitaur for `minitaur-pen`. As seen in the figure, autonomous operation skews the gripper towards the corners in `tabletop-organization`, the peg is stuck around the goal box and minitaur can completely go away from the goal distribution. However, when autonomous algorithms are able to solve the task, the learned policies can be more robust as they are faced with a tougher exploration challenge during training. We visualize this in Figure 4, where we test the final policies learned by oracle RL, FBRL and VaPRL on `tabletop-organization` starting from a uniform state distribution instead of the default ones. We observe that the policies learned by VaPRL and FBRL depreciate by 2% and 14.3% respectively, which is much smaller than the 37.4% depreciation of the policy learned by oracle RL, suggesting that autonomous RL can lead to more robust policies.

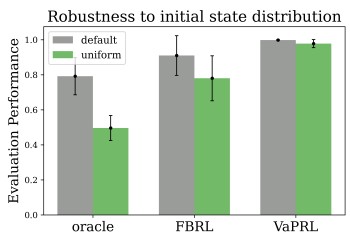

Figure 4: Evaluating policies starting from a uniform initial state distribution. Policies learned via autonomous RL (FBRL and VaPRL) are more robust to initial state distribution than policies learned in oracle RL.

## 7 CONCLUSION

We proposed a formalism and benchmark for autonomous reinforcement learning, including an evaluation of prior state-of-the algorithms with explicit emphasis on autonomy. We present two distinct evaluation settings, which represent different practical use cases for autonomous learning. The main conclusion from our experiments is that existing algorithms generally do not perform well in scenarios that demand autonomy during learning. We also find that exploration challenges, while present in the episodic setting, are greatly exacerbated in the autonomous setting.

While our work focuses on predominantly autonomous settings, there may be task-specific trade-offs between learning speed and the cost of human interventions, and it may indeed be beneficial to provide some human supervision to curtail total training time. How to best provide this supervision (rewards and goal setting, demonstrations, resets etc) while minimizing human cost provides a number of interesting directions for future work. However, we believe that there is a lot of room to improve the autonomous learning algorithms and our work attempts to highlight the importance and challenge of doing so.

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

# A  APPENDIX

## A.1  GOAL-CONDITIONED ARL

This framework can be readily extended to goal-conditioned reinforcement learning (Kaelbling, 1993; Schaul et al., 2015), which is also an area of study covered in some prior works on reset-free reinforcement learning (Sharma et al., 2021). Assuming a goal space $\mathcal{G}$ and task-distribution $p_g : \mathcal{G} \mapsto \mathbb{R}_{\geq 0}$, assume that the algorithm $\mathbb{A} : \{s_i, a_i, s_{i+1}\}_{i=0}^{t-1} \mapsto (a_t, \pi_t)$, where $a_t \in \mathcal{A}$ and $\pi_t : \mathcal{S} \times \mathcal{A} \times \mathcal{G} \mapsto \mathbb{R}_{\geq 0}$. Equation 1 can be redefined as follows:

$$J_D(\pi) = \mathbb{E}_{g \sim p_g, s_0 \sim \rho, a_t \sim \pi(\cdot|s_t, g), s_{t+1} \sim p(\cdot|s_t, a_t)} \Big[ \sum_{t=0}^{\infty} \gamma^t r(s_t, a_t, g) \Big] \tag{3}$$

Additionally, we will assume that the algorithm has access to a set of samples $g_i \sim p(g)$ from the goal distribution. The definitions for deployed policy evaluation and continuing policy evaluation remains the same.

## A.2  REVERSIBILITY AND NON-ERGODIC MDPS

We expand on the discussion of reversibility in 4.2. We also discuss how we can deal with non-ergodic MDPs by augmenting the action space in MDP $\mathcal{M}_T$ with calls for extrinsic interventions.

**Definition 3** (*Ergodic MDPs*). A MDP is considered ergodic if for all states $s_a, s_b \in \mathcal{S}, \exists \pi$ such that $\pi(s_b) > 0$, where $\pi(s) = (1 - \gamma) \sum_{t=0}^{\infty} \gamma^t p(s_t = s \mid s_0 = s_a, \pi)$ denotes discounted state distribution induced by the policy $\pi$ starting from the state $s_0 = s_a$ (Moldovan & Abbeel, 2012). A policy that assigns a non-zero probability to all actions on its support $\mathcal{A}$ ensures that all states in $\mathcal{S}$ are visited in the limit for ergodic MDPs, satisfying the condition above.

We adapt the ARL framework to develop learning algorithms for non-ergodic MDPs. In particular, we introduce a mechanism below for the agent to call for an intervention in the MDP $\mathcal{M}_T$. Our goal here is to show that our described ARL framework is general enough to include cases where a human is asked for help in irreversible states. Consider an augmented state space $\mathcal{S}^+ = \mathcal{S} \cup \mathbb{R}_{\geq 0}$ and an augmented action space $\mathcal{A}^+ = \mathcal{A} \cup \mathcal{A}_h$, where $\mathcal{A}_h$ denotes the action of asking for help via human intervention. A state $s^+ \in \mathcal{S}^+$ can be written as a state $s \in \mathcal{S}$, and $h \in \mathbb{R}_{\geq 0}$ which denotes the remaining budget for interventions into the training. The budget $h$ is initialized to $h_{max}$ when the training begins. The intervention can be requested by the agent itself using an action $a \in \mathcal{A}_h$ or it can enforced by the environment (for example, if the agent reaches certain detectable irreversible states in the environment and requests a human to bring it back into the reversible set of states). For an action $a \in \mathcal{A}_h$, the environment transitions from $(s, h) \to (s', h - c(s, a))$, where the next state $s'$ depends on the nature of the requested intervention $a$ and $c(s, a) : \mathcal{S} \times \mathcal{A}_h \mapsto \mathbb{R}_{\geq 0}$ denotes the cost of intervention. A similar transition occurs when the environment enforces the human intervention. When the cost exceeds the remaining budget, that is $h - c(s, a) \leq 0$, the environment transitions into an absorbing state $s_\emptyset$, and the training is terminated. We retain the definitions introduced in Sec 4.1.1 and 4.1.2 for evaluation protocols.[3] In this way, we see that we can actually encompass the setting of non-ergodic MDPs as well, with some reparameterization of the MDP.

## A.3  RELATING PRIOR WORKS TO ARL

In this section, we connect the settings in prior works to our proposed autonomous reinforcement learning formalism. First, consider the typical reinforcement learning approach: Algorithm $\mathbb{A}$ assigns the rewards to transitions using $r(s, a)$, learns a policy $\pi$ and outputs $\pi_t = \pi$ and $a_t \sim \pi$ (possibly adding noise to the action). The algorithm exclusively optimizes the reward function $r$ throughout training.

Reconsider the door closing example: the agent needs to practice closing the door repeatedly, which requires opening the door repeatedly. However, if we optimize $\pi$ to maximize $r$ over its lifetime, it will never be incentivized to open the door to practice closing it again. In theory, assuming an ergodic MDP and that the exploratory actions have support over all actions, the agent will open the door given enough time, and the agent will practice closing it again. However, in practice, this can be quite an inefficient strategy to rely on and thus, prior reset-free reinforcement learning algorithms consider other strategies for exploration. To understand current work, we introduce the notion of

---

[3]The action space $\mathcal{A}_h$ is not available in $\mathcal{M}$, only in $\mathcal{M}_T$. This discrepancy should be considered when parameterizing $\pi$.

a surrogate reward function $\tilde{r}_t : \mathcal{S} \times \mathcal{A} \mapsto \mathbb{R}$. At every time step $t$, $\mathbb{A}$ outputs $a_t \sim \pi_e$ and $\pi_t$ for evaluation, where $\pi_e$ optimizes $\tilde{r}_t$ over the transitions seen till time $t - 1$ [4]. Prior works on reset-free reinforcement learning can be encapsulated within this framework as different choices for surrogate reward function $\tilde{r}_t$. Some pertinent examples: Assuming $r_\rho$ is some reward function designed that shifts the agent's state distribution towards initial state distribution $\rho$, alternating $\tilde{r}_t = r$ and $\tilde{r}_t = r_\rho$ for a fixed number of environment steps recovers the forward-backward reinforcement learning algorithms proposed by Han et al. (2015); Eysenbach et al. (2017). Similarly, R3L (Zhu et al., 2020) can be understood as alternating between a perturbation controller optimizing a state novelty reward and the forward controller optimizing the task reward $r$. Recent work on using multi-task learning for reset-free reinforcement learning (Gupta et al., 2021) can be understood as choosing $\tilde{r}_t(s_t, a_t) = \sum_{k=1}^{K} r_k(s_t, a_t)\mathbb{I}[s_t \in \mathcal{S}_k]$ such that $\mathcal{S}_1, \ldots \mathcal{S}_K$ is a partition of the state space $\mathcal{S}$ and only reward function $r_k$ is active in the subset $\mathcal{S}_k$. Assuming the goal-conditioned autonomous reinforcement learning framework, the recently proposed algorithm VaPRL (Sharma et al., 2021) can be understood as creating a curriculum of goals $g_t$ such that at every step, the action $a_t \sim \pi(\cdot \mid s_t, g_t)$. The curriculum simplifies the task for the agent, bootstrapping on the success of easier tasks to efficiently improve $J_D(\pi)$.

## A.4 Environment Descriptions and Reward Functions

**Tabletop-Organization.** The Tabletop-Organization task is a diagnostics object manipulation task proposed by Sharma et al. (2021). The observation space is a 12 dimensional vector consisting of the object position, gripper position, gripper state, and the current goal. The action space is 3-D action that consists of a 2-D position delta and an automated gripper that can attach to the mug if the gripper is close enough to the object. The reward function is a sparse indicator function, $r(s, g) = \mathbb{1}(\|s - g\|_2 \leq 0.2)$. The agent is provided with 12 forward and 12 backward demonstrations, 3 for each of the 4 goal positions.

**Sawyer-Door.** The Sawyer Door environment consists of a Sawyer robot with a 12 dimension observation space consisting of 3-D end effector position, 3-D door position, gripper state and desired goal. The action space is a 4-D action space that consisting of a 3-D end effector control and normalized gripper torque. Let $s_d$ be dimensions of the observation corresponding to the door state, and $g$ be the corresponding goal. The reward function is a sparse indicator function $r(s, g) = \mathbb{1}(\|s_d - g\|_2 \leq 0.08)$. The agent is provided with a 5 forward and 5 backward demos.

**Sawyer-Peg.** The Sawyer-Peg environment shares observation and action space as the Sawyer-Door environment. Let $s_p$ be the state of the peg and $g$ be the corresponding goal. The reward function is a sparse indicator function $r(s, g) = \mathbb{1}(\|s - g\|_2 \leq 0.05)$. The agent is provided with 10 forward and 10 backward demonstrations for this task.

**Franka-Kitchen.** The Franka-Kitchen environment consists of a 9-DoF Franka robot with an array of objects (microwave, two distinct burner, door) represented by a 14 dimensional vector. The reward function is composed of a dense reward that is a sum of the euclidean distance between the goal position of the arm and the current state plus shaped reward per object as described in Gupta et al. (2019). No demonstrations are provided for this task.

**DHand-LightBulb.** The DHand is a 4 fingered robot (16-DoF) mounted on a 6-DoF Sawyer Arm. The observation space of the DHand consists of a 30 dimensional observation and corresponding goal state. The observation is composed of a 16 dimensional hand position, 7 dimensional arm position, 3 dimension object position, 3 dimensional euler angle and a 6 dimensional vector representing the dimensional wise distance to between objects in the environment. The action space is a position delta over the combined 22 DoF of the robot. No demonstrations are provided for these tasks.

**Minitaur-Pen.** The Minitaur pen's observation space is the joint positions of its 8 links, their corresponding velocity, current torque, quaternion of its base position, and goal location in the pen. The action space is a 8 dimensional PD target. Let $s_b$ be the 2-D position of the agent, and $g$ be the corresponding goal. Let $s_t$ be the current torques on the agent, and $s_v$ be their velocities The reward for the agent is a dense reward $r(s, a) = -2.0 \cdot \|s_b - g\| + 0.02 \cdot \|s_v \cdot s_t\|$. No demonstrations are provided for these tasks.

---

[4] This can also be captured in a multi-task reinforcement learning framework, where $\pi_e, \pi_t$ are the same policy with different task variables as input.

## A.5 ALGORITHMS

We use soft actor-critic (Haarnoja et al., 2018b) as the base algorithm for our experiments in this paper. When available, the demonstrations are added to the replay buffer at the beginning of training. Further details on the parameters of the environments and algorithms are reported in Tables A.5. The implementations will be open-sourced and implementation details can be found there.

| Hyperparameter | Value |
|---|---|
| Actor-critic architecture | fully connected(256, 256) |
| Nonlinearity | ReLU |
| RND architecture | fully connected(256, 256, 512) |
| RND Gamma | 0.99 |
| Optimizer | Adam |
| Learning rate | 3e-4 |
| $\gamma$ | 0.99 |
| Target update $\tau$ | 0.005 |
| Target update period | 1 |
| Batch size | 256 |
| Classifier batch size | 128 |
| Initial collect steps | $10^3$ |
| Collect steps per iteration | 1 |
| Reward scale | 1 |
| Min log std | -20 |
| Max log std | 2 |

Table 3: Shared algorithm parameters.

| Environment | Training Horizon ($H_T$) | Evaluation Horizon ($H_E$) | Replay Buffer Capacity |
|---|---|---|---|
| Tabletop-Organization | 200,000 | 200 | 20,000,000 |
| Sawyer-Door | 200,000 | 300 | 20,000,000 |
| Sawyer-Peg | 100,000 | 200 | 20,000,000 |
| Franka-Kitchen | 100,000 | 400 | 10,000,000 |
| DHand-Lightbulb | 400,000 | 400 | 10,000,000 |
| Minitaur-Pen | 100,000 | 1000 | 10,000,000 |

Table 4: Environment specific parameters, including the training horizon (i.e. how frequently an intervention is provided), the evaluation horizon, and the replay buffer capacity.

## A.6 EVALUATION CURVES

In this section, we plot deployed policy evaluation and continuing policy evaluation curves for different algorithms and different environments:

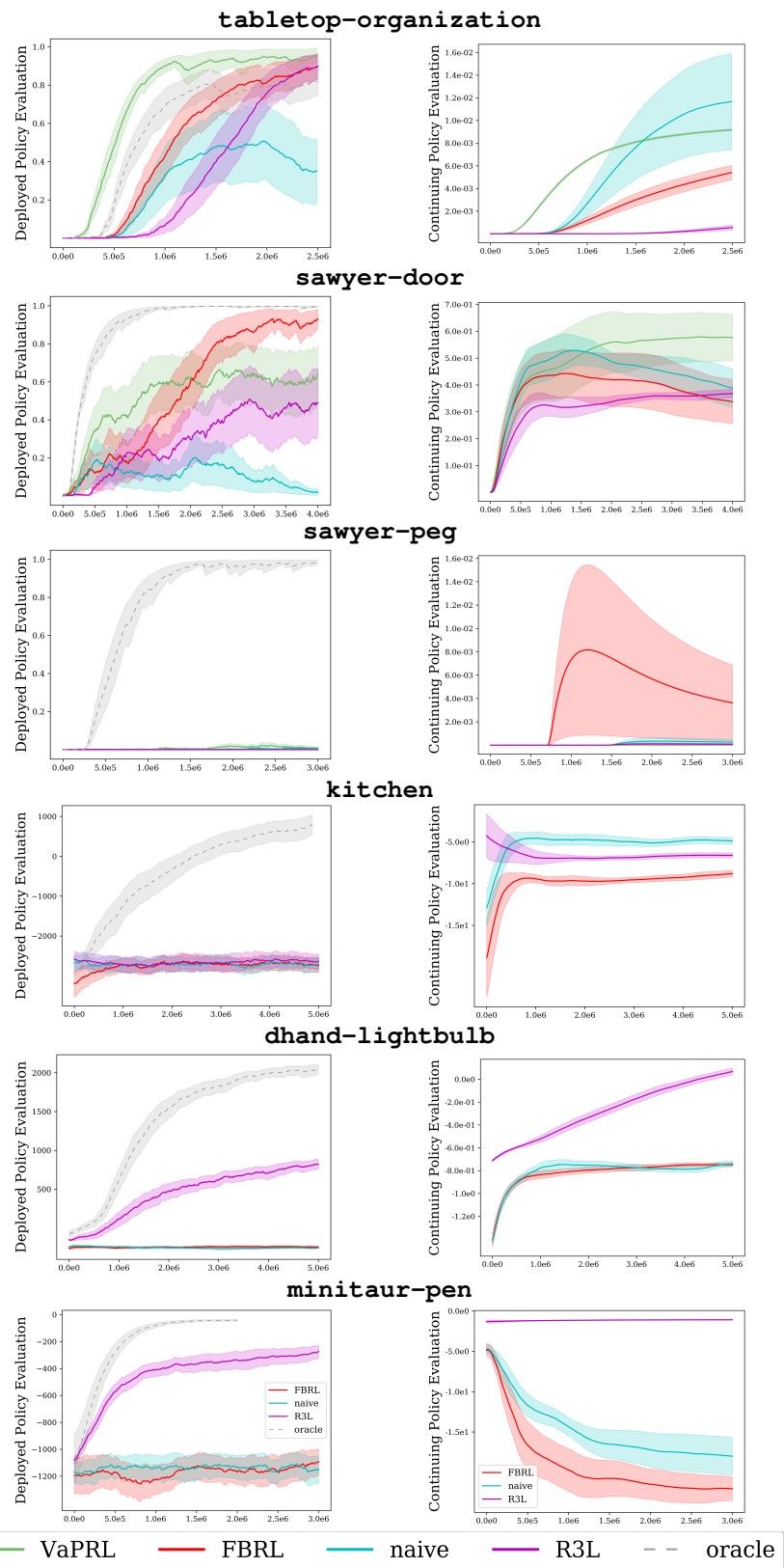

Figure 5: Deployed Policy Evaluation and Continuing Policy Evaluation per environment. Results and averaged over 5 seeds. Shaded regions denote 95% confidence bounds.

