# OpenReview forum: "Autonomous Reinforcement Learning: Formalism and Benchmarking"
_ICLR.cc/2022/Conference — ICLR 2022 Poster_

### Official Review · Reviewer_AL8i · 2021-10-30

**Correctness:** 2
**Technical Novelty And Significance:** 1
**Empirical Novelty And Significance:** 2
**Recommendation:** 3
**Confidence:** 4

**Main Review:**

Even if I always commend papers that propose benchmarks and evaluation framework, I found this work quite problematic. First of all, the authors use the expression autonomous reinforcement learning, which is per se already quite "charged". In fact, the concept of autonomy is rather well defined in our community. It is unclear why the behavior described in this work is considered "autonomous". Usually, the concept of autonomy is defined with respect to the concept of human intervention in setting goals, etc. The authors do not consider this type of autonomy. Instead, they differentiate systems according to learning based on episodes or not. It seems to me that the actual classification used by the authors is not really distant from the existing separation between continual and non continual learning.

The authors also discuss a distinction between systems that are trained using episodic training and then deployed as continual learning systems, but the reviewer believes that it is not really necessary to devise new benchmarks, etc. since existing ones can be considered sufficient to deal with these situations.

The definition of deployed policy evaluation (Definition 1) is rather unclear. How are you going to derive that? It seems to me that in practical terms, the calculation of such a value is not possible.

Definition 2 ("continual policy evaluation") appears to me as the classic definition of a RL problem in a non-episodic case.

The reviewer also does not see the need of introducing the concept of "irreversible states". At the end of the day, a situation where an agent is stuck in a certain situation might happen. The reviewer is not sure why this case should be considered separately. The reviewer also wonders if there is any specific requirement for this assumption.

The models used in the benchmarks are not novel (see Section 5), but they are essentially existing ones. The contributions of the authors with respect these existing environments is unclear.

The evaluation appears to be a sort of standard analysis of the performance of the environments under some specific conditions. The reviewer was not able to see a novel contribution in it.

Minor points:

- The authors say: "Current episodic settings typically provide an environment reset every 100 to 1000 steps, corresponding to ε ∈ (1e-3, 1e-2) and an autonomous operation time of typically a few seconds to few minutes depending on the environment.". I am not sure about this claim of the authors. The reset really depends on the actual experiments.

- The authors say: "In essence, the policy output from the algorithm A trained in the autonomous training setting should do well on the deployment setting as quickly as possible." I am not sure about this - I think it really depends on the environment, if it is stationary, etc. Also, the authors are assuming here that learning happens at deployment time as well and this is not always the case.


**Summary Of The Paper:**

The paper describes a framework for benchmarking episodic and non-episodic algorithms. The authors provide both a mathematical formalisation and a description of an experimental framework composed of existing environments. It is quite difficult to see clear contributions in this paper since the theoretical framework does not really introduce novel concepts and the experimental framework is essentially a collection of existing environments.

**Summary Of The Review:**

The contributions are rather limited in my opinion since the theoretical framework essentially essentially proposes re-definitions of existing concepts. The experimental framework is composed of existing benchmarks so it is difficult to see a contributions there.

---

> ### Author Response · Authors · 2021-11-16
> **Response to Reviewer AL8i (1/2)**
>
> Thanks for the review! We have made multiple revisions based on your suggestions, and we hope that this improves the overall clarity of the paper. We want to clarify the context for this work to better situate it, then we address the specific concerns raised in the review.
>
> We want embodied agents to be able to learn with minimal interventions required for training. There are several challenges that come along the way such as lack of exact rewards, non-stationarity in tasks and dynamics. However, there is another major discrepancy between this vision of agents learning and the way our agents learn: episodic lifetimes. More specifically, human supervision in the form of environment resets is implicitly required for episodic learning. Removing this unstated constraint poses a challenging problem that once solved would be a major step towards autonomous learning agents.
>
> We have rewritten the related work (Section 2) to better recognize that human interventions are also required in reward design and goal setting, to highlight prior work in the area of reward/goal specification, and to clarify our contribution.
>
> > It seems to me that the actual classification used by the authors is not really distant from the existing separation between continual and non continual learning
>
> As you note, the motivation of this work fits with the broader goals of continual learning. But, continual learning research has predominantly focused on non-stationary task distributions and transfer of knowledge between tasks and domains [1, 2, 3, 4]. Autonomous RL (i.e., non-episodic lifetimes) is a hard learning problem, even when the task-distribution and dynamics are stationary (see Section 6.1). In fact, the prior works in continual RL assume episodic lifetimes [1, 2, 3, 4], which implicitly necessitates frequent human intervention.
>
> We apologize for the lack of clarity and we have revised our draft in both the related work (Section 2) and formalism section (4.1) to specifically state the differences highlighted above. Please let us know if this helps to address this concern.
>
> > It is unclear why the behavior described in this work is considered "autonomous". Usually the concept of autonomy is defined with respect to the concept of human intervention in setting goals, etc. The authors do not consider this kind of autonomy.
>
> Thanks for this comment. We agree with the notion of that autonomy encompasses other areas of human intervention, and we have revised the paper in multiple sections (Related Work - Sec 2, Formalism - Sec 4.1, Conclusion - Sec 7) to clarify that goal setting itself requires human intervention and discuss literature in self-supervised goal setting. We would like to emphasize that the intervention requirement implicit in episodic learning is both significant and underemphasized. Episodic learning requires interventions at the end of every trajectory (which in the real world could happen every minute), whereas selecting goal distributions or crafting reward functions can in principle be only done once before training. We believe removing this requirement gives us the biggest increment in autonomy. Other areas of work, such as safe RL, continual RL, and unsupervised/self-supervised RL, also reduce human intervention into learning, but unlike them, autonomous RL also lacks a formal definition or standardized benchmarks (which our work tries to ameliorate).
>
> > The authors also discuss a distinction between systems that are trained using episodic training and then deployed as continual learning systems,
>
> We want to clarify that ARL, even in the deployed setting, never assumes any episodic training. The agents always learn in a non-episodic setting. ARL proposes two different evaluation schemes: deployment and continuing. Continuing policy evaluation monitors the reward collected over the agent’s lifetime whereas deployed policy evaluation computes the return of hypothetical policy rollouts from the initial state distribution if it were deployed. This isn’t required for training, and training continues without human interventions. We apologize for the confusion and we have revised the text in Section 4.1.1 to be clearer about this.
>
> > but the reviewer believes that it is not really necessary to devise new benchmarks, etc. since existing ones can be considered sufficient to deal with these situations.
>
> We believe that current widely used benchmarks are not sufficient to evaluate such situations, as evidenced by the fact that, in our evaluations, prior methods that were designed to perform on current benchmarks generally perform quite poorly on the proposed ARL benchmarks. Of course, we would be happy to cite, discuss, and compare to other benchmarks that you believe may be relevant, and would appreciate any pointers to relevant benchmarks we may have missed, but we are not aware of other benchmarks that cover the autonomous RL setting.

---

> > ### Author Response · Authors · 2021-11-16
> > **Response to Reviewer AL8i (2/2)**
> >
> > > The definition of deployed policy evaluation (Definition 1) is rather unclear. How are you going to derive that? It seems to me that in practical terms, the calculation of such a value is not possible.
> >
> > We revised the text in Section 4.1.1 to clarify the deployed evaluation metric. We also agree that the quantity itself cannot be computed exactly but we discuss a practical approximation for the quantity is discussed in paragraph *Evaluation Metrics* (Section 6.2). In practice, $J_D(\pi)$ is approximated as the return over a fixed horizon (as is conventionally done for standard RL which assumes an infinite horizon), and we compute $J_D(\pi_t)$ intermittently (in our experiments, every 10000 steps). A better algorithm would have a higher area under the curve when $J_D(\pi_t)$ is plotted versus $t$. Please let us know if this clarification helps, and we are happy to further discuss this and make appropriate revisions to enhance clarity.
> >
> > > The reviewer also does not see the need to introduce the concept of "irreversible states"... if there is any specific requirement for this assumption.
> >
> > We agree that the concept of irreversible states is an orthogonal consideration to the main message of the paper. We have shifted majority of the discussion from Section 4.2 to Appendix A.2. Please let us know if this refactoring helps.
> >
> > > The models used in the benchmarks are not novel (see Section 5)...
> >
> > We recognize this in text [Introduction, Paragraph 4, Line 2-3]. Our benchmarking effort focused on standardizing ARL and evaluating existing algorithms on existing environments, but under the constraints of ARL. In fact, showing how the current algorithms perform poorly on current tasks reflects the challenge of ARL and the importance of choosing appropriate training/evaluation schemes. As a concrete example, the Sawyer-Door problem from Meta-world [5] is reset after every 500 steps in the environment. In our benchmark EARL, the environment is reset after every 200,000 steps during training. This is a 400x difference that necessitates development of new algorithms (Section 6.1), and substantially reduces the number of human interventions for environment resets.
> >
> > > The evaluation appears to be a standard analysis …
> >
> > We believe our proposed problem setting of ARL is novel, and we are not aware of a similar evaluation/benchmarking under our proposed constraints. We would be happy to discuss and cite any works that do similar analysis to ours, and would appreciate pointers to such an analysis as we may have missed it.
> >
> > Minor points:
> > - We agree the episode length depends on the specific task. However most contemporary benchmarks reset the environment in less than 1000 steps [5, 6, 7].
> > - We want to clarify that there is no learning happening at deployment. The algorithm $\mathbb{A}$ should output a “good” policy for deployment as soon as possible (the deployment itself is stationary).
> >
> > We will be happy to update the manuscript to make any more clarifications, and we hope the discussion here provides more context to understand our work. Please let us know how we can further improve our manuscript.
> >
> > [1] Towards Continual Reinforcement Learning: A Review and Perspectives. Khetarpal et al. (https://arxiv.org/pdf/2012.13490.pdf)
> >
> > [2] Progressive Neural Networks. Rusu et al. (https://arxiv.org/abs/1606.04671)
> >
> > [3] Overcoming Catastrophic Forgetting in Neural Networks. Kirkpatrick et al. (https://arxiv.org/abs/1612.00796)
> >
> > [4] Continual World: A Robotic Benchmark for Continual Reinforcement Learning. Wołczyk et al. NeurIPS 2021.
> >
> > [5] Meta-World: A Benchmark and Evaluation for Multi-Task and Meta Reinforcement Learning. Yu et al.  CoRL, 2019
> >
> > [6] OpenAI Gym. Brockman et al.
> >
> > [7] DeepMind Control Suite. Tassa et al.

---

### Official Review · Reviewer_sj9u · 2021-11-01

**Correctness:** 4
**Technical Novelty And Significance:** 3
**Empirical Novelty And Significance:** 3
**Recommendation:** 8
**Confidence:** 4

**Main Review:**

Pros:

Excellent example of carefully chosen augmentations of a standard problem formulation in order to factor in hidden costs with substantial practical implications in several experimental settings of interest, such as real-world robotics experiments.

Well designed toy domains which emulate some such difficulties for quick experimentation in a comparable way.


Cons:

It is not clear whether the proposed methodology really deals with issues of environment resets in the fullness of time. Will following this methodology result in much fewer resets and superior learning if tasked to achieve the same level of performance as expensive and impractical full episodic reset methodology, at least for tasks of interest with real-world robots? While algorithms developed for ARL may indeed require fewer interventions in a fixed amount of experiment time, this does not mean that convergence to an optimal policy will not take considerably more experimentation time than episodic reset approaches. If total experimentation time is also a factor, then the trade-off of autonomous operation in the real-world vs. cost of human resets seems to be still an open problem in need of task specific trade-offs.

Perhaps there is no choice but to resort to ARL, but it is imho not clear that ARL doesn’t introduce learning issues of its own, since learning from biased data streams is known to be particularly challenging and empirically leads to suboptimal learning, as per the continual learning literature. This view is not necessarily invalidated by experimental evidence presented in the paper, at least as far as I can tell.

More discussion and standardization may be needed in order to reduce variance of ARL experiments when reporting results. Without the aid of environment resets, ARL data sampling is even more contingent on learned policies and learning dynamics of particular algorithms and function approximators. Hence, standard convergence assumptions and hopes may be even more ephemeral, especially with non-linear function approximation. Also, sensitivity to implementation details may be increased, e.g. hyperparameters, software libraries and their default settings, even model initializations, etc.


**Summary Of The Paper:**

This paper highlights the important concept of autonomous RL, inspired by real-world scaling constraints of robot learning experiments. Hiding the cost of environmental resets is unrealistic and hinders progress in embodied RL. This incentivises the search for algorithms which score better trade-offs between expensive environment resets and average return of learned policies, using the two definitions given in the paper. Clear conceptual problem formulations are given which reproduce challenges observed during real-world RL experiments with robots, but in tractable settings using simulation.

**Summary Of The Review:**

This paper gives two convincing definitions of autonomous reinforcement learning and introduces clear evaluation methodologies accessible for the community. Relevant baseline results are provided, such that the proposed benchmarks are good starting points for future research. While some conceptual challenges linger, this could be resolved by followup works.

---

> ### Author Response · Authors · 2021-11-15
> **Response to Reviewer sj9u**
>
> Thanks a lot for your thoughtful review! We largely agree with the points that you have raised in the review. We provide some thoughts here:
>
> > The trade-off of autonomous operation vs cost of human resets seems to be still an open problem in need of a task-specific trade-offs
>
> We agree that there may be a trade-off between how often a human intervenes and total experimentation time, and this may be task dependent. We also believe that there is a lot of room to develop better algorithms and methodologies conducive for autonomous learning and that such development may make the trade-off clearer. We have added discussion to Section 4.3 and Section 7 to reflect your comments.
>
> > learning from biased data streams is known to be particularly challenging and empirically leads to suboptimal learning, as per the continual learning literature
>
> We agree that autonomous operation creates a harder learning challenge as the data distribution can be biased. Related discussion can be found in Section 6.3 where we show the agent’s exploration leads to broader state visitation. While learning can be challenging, on the plus side, autonomous learning forces the agent to correct its mistakes [in contrast to an environment reset resolving it], potentially leading to more robust learning suitable for the real-world [also discussed in Section 6.3].
> > More discussion and standardization .. sensitivity to implementation details ..
>
> This is indeed a valid concern. We plan to open-source the code for both the benchmarked baselines and the environments, which we hope will provide some standardization. We have additionally provided implementation details and hyperparameters for all of the baselines in the Appendix.

---

> > ### Comment · Reviewer_sj9u · 2021-11-25
> > **Thank you for addressing the points raised**
> >
> > Imho, the paper makes a convincing argument for ARL being a distinct enough paradigm of practical importance to deserve a standardized methodology for future work, and it delivers exactly that. While conceptually clear, the standard RL methodology has largely not yet proven generally practical for real-world control learning, at least with high-dimensional inputs and complex actuators. This paper isolates one problematic assumption in the way of such progress, the environment reset problem, and provides the groundwork for progress, i.e. baselines, clear metrics, etc. I believe the community is much better off with this paper published, since prior works don't seem to have used compatible methodologies.

---

### Official Review · Reviewer_omLm · 2021-11-03

**Correctness:** 4
**Technical Novelty And Significance:** 2
**Empirical Novelty And Significance:** 1
**Recommendation:** 5
**Confidence:** 4

**Main Review:**

The strength of this paper is that they clearly argue that the studies on continual learning are important and give a clear notion about ARL.
The weakness of this paper is the lack of theoretical novelty and new proposal except for the argument around ARL and its benchmark tasks.


**Summary Of The Paper:**

This paper describes a new framework called Autonomous Reinforcement Learning (ARL) to facilitate studies on continual real-world embodied learning, such as that performed by humans and animals. Most reinforcement learning (RL) studies use episodic benchmark tasks. The paper argues that conducting studies on non-episodic tasks is crucial. After defining ARL, the authors introduce simulated benchmarks.

**Summary Of The Review:**

I totally agree with the authors' argument about the importance of continual reinforcement learning. I also agree that the RL culture that makes agents' life "episodic" tends to prevent us from tackling the problem. I also think the "non-episodic" feature is one of the important ones, although the real-world embodied learning, such as that performed by humans and animals, has many other essential features, e.g., the reward function is not unique and environmental dynamics are changing over time even in an episode,
Therefore, I like the argument of the authors, especially in the Introduction.

However, it's questionable if the paper has a sufficient paper as an ICLR conference paper though it has great importance in its aspect of "position paper."

1) Theoretical contribution around ARL.
To my understanding, the theoretical notion of ARL is not new. The general and classical framework of RL does not necessarily have the assumption of episodic segmentation. Therefore, the novelty of the authors' proposal about ARL is limited in a theoretical viewpoint though it is quite meaningful in the "cultural" viewpoint.

2) Benchmarks.
The authors described some benchmarks. However, to my understanding, even though they showed the pre-existing RL algorithms underperformed in the ARL settings, they are not providing a suitable task for developing ARL algorithms. This is because they only showed the pre-existing methods tend to underperform in the ARL settings.
Some more constructive proposal is expected.

The paper is very good from the viewpoint of position/short journal paper. However, I am not sure if the paper is suitable for the ICLR conference paper.

---

> ### Author Response · Authors · 2021-11-15
> **Response to Reviewer omLm**
>
> Thank you for your review and feedback! We have included a detailed response to your comments below. Specifically with regards to the suitability of this paper for ICLR, we would just like to point out that a number of papers with similar contributions have been accepted to multiple comparable ML venues (see below). Please let us know if this helps to address your concerns. We are happy to continue discussing if you have further comments/suggestions related to anything in the paper.
>
> > The paper is very good from the viewpoint of position/short journal paper. However, I am not sure if the paper is suitable for the ICLR conference paper
>
> Several recent papers [1, 2, 3, 4, 5, 6] with similar nature of contributions were accepted to comparable venues, i.e. NeurIPS, ICML, and ICLR, proposing benchmarks on continual RL, off-policy evaluation, distribution shifts, and other topics. Benchmarks have been catalysts for research, development, and analysis of algorithms. Not only could autonomous RL benefit from the formalism and standardized evaluation for development of algorithms, but we hope that our paper serves as a call to action for building autonomous embodied agents.
>
> > To my understanding, the theoretical notion of ARL is not new.
>
> The continuing ARL setting closely follows the standard RL formalism, but the notion that such a formulation can be adapted effectively for autonomous learning runs counter to the approaches proposed in other recent works [7, 8, 9], which does not address this setting. The formalization of the deployment ARL setting is also new; although these prior works do suggest various forms of forward-backward controllers, to our knowledge they do not formalize the precise and general problem statement that encompasses such problems, nor recognize that it corresponds to a transfer learning problem.
>
> > The general and classical framework of RL does not necessarily have the assumption of episodic segmentation
>
> We agree that the general classical framework of RL does not need the episodic segmentation. However, as our empirical analysis shows, the algorithms developed on current benchmarks with episodic lifetimes do not translate to the ARL setting, which is what motivated the creation of this benchmark. This gap between algorithms developed against episodic benchmarks also greatly impacts the applicability of the algorithms to real world settings, which may require autonomous RL.
>
> > Some more constructive proposal is expected.
>
> The paper provides a hypothesis/analysis for why current algorithms are insufficient in Section 6.3, which can potentially be used to develop better algorithms. In particular, algorithms can pursue more conservative forms of exploration. For example, a recent work [10] factors in reversibility of actions to be more conservative in exploration. We believe that emphasizing the importance of this problem statement, which has been largely neglected in the recent history of RL research, is important in order to move toward algorithms that are applicable for real-world learning, and we hope that our benchmarks will also serve as a constructive step to facilitate this.
>
> [1] Wilds: A Benchmark of in-the-Wild Distribution Shifts. Koh & Sagawa et al. ICML 2021.
>
> [2] Leveraging Procedural Generation to Benchmark Reinforcement Learning. Cobbe et al. ICML 2020.
>
> [3] Continual World: A Robotic Benchmark for Continual Reinforcement Learning. Wołczyk et al. NeurIPS 2021.
>
> [4] Benchmarks for Deep Off-Policy Evaluation. Fu et al. ICLR 2021.
>
> [5] BREEDS: Benchmarks for Subpopulation Shift. Santurkar et al. ICLR 2021.
>
> [6] IsarStep: a Benchmark for High-level Mathematical Reasoning. ICLR 2021.
>
> [7] Leave no Trace: Learning to Reset for Safe and Autonomous Reinforcement Learning. Eysenbach et al. NeurIPS 2017.
>
> [8] The Ingredients of Real World Robotic Reinforcement Learning. Zhu et al. ICLR 2020.
>
> [9] Autonomous Reinforcement Learning via Subgoal Curricula. Sharma et al. NeurIPS 2021.
>
> [10] There Is No Turning Back: A Self-Supervised Approach for Reversibility-Aware Reinforcement Learning. Grinsztajn et al. NeurIPS 2021.

---

> > ### Comment · Reviewer_AL8i · 2021-11-28
> > **Thanks for the replies, but the contents of this work are still problematic in my opinion**
> >
> > My concern about the actual framing of the problem against existing definition of continual learning vs autonomous RL are still there.
> >
> > In fact the authors say that episodic lifetimes "implicitly necessitates human intervention". It does not seem the case in general - in theory this might just automatic.
> >
> > The use of the term "autonomy" in the text is still not standard with respect to the literature. The need of calling this RL as autonomous is still not convincing for me.
> >
> > Again, in terms of benchmarks it is not clear to me why you need new benchmarks given the fact that these appear as limited cases of existing ones.
> >
> > Re definition 1, it seems to me that you are then consider a finite case with that new definition? The general definition does not have a real mapping in practice, which appears one of the goals of the work of the authors.
> >
> > Re the evaluation, the authors say that the novelty is about the study "under constraints". In my opinion, these are instead reframing of cases (which might be considered limit cases), but they are part of the existing paradigms.
> >
> > I am not in a position to accept this paper for publication.

---

### Official Review · Reviewer_C7Cj · 2021-11-06

**Correctness:** 4
**Technical Novelty And Significance:** 2
**Empirical Novelty And Significance:** 3
**Recommendation:** 8
**Confidence:** 4

**Main Review:**

Strengths
-------------

- The paper provides a nice formalization of an important problem (that of Autonomous Reinforcement Learning)
- I like the distinction into "Deployment Setting" and the "Continuing Setting". Usually people when hearing of ARL, they have in mind the "Continuing Setting", but the "Deployment Setting" is also very useful. More importantly, they are different problems. In any case, the distinction can make things easier to talk about and compare methods/ideas.
- Nice benchmark environments capturing a wide variety of tasks/scenarios
- Thorough analysis providing useful insights (mostly validating our intuition, but this is important!)

Weaknesses
------------------

- The ARL problem is strongly related to the autonomous skill discovery problem (see for example {1,2,3}). For instance, without proper exploration and skill distillation (either automatic or with some heuristic), ARL is not really a solvable problem without many resets. I would definitely like to see a nice discussion about this and other similarities. This would make the paper complete.


References
-----------------
{1}: Campos, V., Trott, A., Xiong, C., Socher, R., Giró-i-Nieto, X. and Torres, J., 2020, November. Explore, discover and learn: Unsupervised discovery of state-covering skills. In International Conference on Machine Learning (pp. 1317-1327). PMLR.

{2}: Gregor, K., Rezende, D.J. and Wierstra, D., 2016. Variational intrinsic control. arXiv preprint arXiv:1611.07507.

{3}: Pong, V.H., Dalal, M., Lin, S., Nair, A., Bahl, S. and Levine, S., 2020, November. Skew-fit: State-covering self-supervised reinforcement learning. In International Conference on Machine Learning. PMLR.



**Summary Of The Paper:**

The manuscript presents a concrete formalization of the "Autonomous Reinforcement Learning" (ARL) problem: this problem refers to a scenario where the agents has no (or very few) resets during learning of the task(s). The paper also unifies previous works into two evaluation scenarios: (a) Deployed Policy Evaluation, and (b) Continuing Policy Evaluation. The manuscript, also, provides a benchmark: "Environments for Autonomous Reinforcement Learning (EARL)". This benchmark contains diverse environments from simple manipulation tasks to locomotion and complex manipulation. Using this benchmark the authors provide an analysis of the crucial factors that affect the performance of current state-of-the-art algorithms for the ARL problem.

The main contributions of the paper are:

- Concrete formalization of the "Autonomous Reinforcement Learning" (ARL) problem
- A novel benchmark for ARL with diverse tasks
- Novel evaluation settings (Deployment Setting and Continuing Setting) for ARL
- Insights on the important factors for effective ARL


**Summary Of The Review:**

The manuscript was a nice and interesting read. The formalism is solid, the benchmark is going to be useful to many researchers and the insights are interesting and although somewhat expected from intuition, it is good to see them result from experiments and written down nicely.

---

> ### Author Response · Authors · 2021-11-15
> **Response to Reviewer C7Cj**
>
> Thanks for your thoughtful comments! We hope the formalism and the benchmark lays the foundation of developing autonomous embodied agents.
>
> >The ARL problem is strongly related to the autonomous skill discovery problem
>
> We agree that works on skill discovery are relevant to the goals of autonomous RL. We have cited and added a discussion around these prior works. Please find the edits (in blue) in the related work section.

---

### Decision · Program_Chairs · 2022-01-20

**Decision:**

Accept (Poster)

**Comment:**

This paper formalizes the setting where an autonomous RL agent operates with zero or very few resets, and provides a novel benchmark for this setting with diverse environments ranging from simple manipulation to complex manipulation/locomotion. The paper then uses this benchmark to analyze current methods and provide insight into those crucial factors that affect performance in this setting. The insights into current methods especially are appreciated. As one reviewer stated, "This paper isolates one problematic assumption in the way of [progress in RL], the environment reset problem, and provides the groundwork for [such] progress, i.e. baselines, clear metrics, etc. I believe the community is much better off with this paper published, since prior works don't seem to have used compatible methodologies."